# Ethyne Functionalized *Meso*-Phenothiazinyl-Phenyl-Porphyrins: Synthesis and Optical Properties of Free Base Versus Protonated Species

**DOI:** 10.3390/molecules25194546

**Published:** 2020-10-04

**Authors:** Eva Molnar, Emese Gál, Luiza Găină, Castelia Cristea, Luminița Silaghi-Dumitrescu

**Affiliations:** Faculty of Chemistry and Chemical Engineering, Babeş-Bolyai University, 11 Arany Janos street, RO-400028 Cluj-Napoca, Romania; molnar_evike@yahoo.com (E.M.); gluiza@chem.ubbcluj.ro (L.G.); lusi@chem.ubbcluj.ro (L.S.-D.)

**Keywords:** phenothiazine, porphyrin, photophysical properties

## Abstract

Synthesis, structural characterization and photophysical properties for a series of new *trans*-A_2_B_2_- and A_3_B-type ethynyl functionalized *meso*-phenothiazinyl-phenyl porphyrin derivatives are described. The new compounds displayed the characteristic porphyrin absorption spectra slightly modified by weak auxochromic effects of the substituents and fluorescence emission in the range of 651–659 nm with 11–25% quantum yields. The changes recorded in the UV-vis absorption spectra in the presence of trifluoroacetic acid (TFA) are consistent with the protonation of the two internal nitrogen atoms of the free-base porphyrin (19 nm bathochromic shift of the strong Soret band and one long wave absorption maxima situated in the range of 665–695 nm). Protonation of the phenothiazine substituents required increased amounts of TFA and produced a distinct hypsochromic shift of the long wave absorption maxima. The density functional theory (DFT) calculations of a porphyrin dication pointed out a saddle-distorted porphyrin ring as the ground-state geometry.

## 1. Introduction

A large variety of scientific articles describing tetrapyrrolic compounds and their metal derivatives were stated in the literature, most of them documenting synthetic procedures [1,2], chemical, photophysical properties and applications as photosensitizers in photodynamic therapy [3,4] including two-photon excitation [5], active materials for organic solar cells [6,7] or light emitting diodes [8,9,10], chemical/biosensors [11,12,13] and biologically active compounds [14,15]. The UV-Vis absorption spectroscopy was widely applied as a suitable analytical procedure for the investigation of the electronic structure of the porphyrins characterized by typical absorption and fluorescence emission bands situated in the visible region (Soret-band positioned at about 350–500 nm generally with molar absorptivity of 10^5^ M^−1^ cm^−1^, and up to four Q-bands at 500–750 nm with lower intensities) closely associated with electronic transitions influenced by the aromatic core electronic distribution [14,16]. The chemical and sensing properties could be tuned by introducing different acceptor/donor moieties on *meso* and/or *β* positions of the porphyrin or by core complexation with transition metals. Disturbances were reported for the characteristic absorption spectra of porphyrin derivatives as a result of different factors such as: conjugation pathway and symmetry [17]. Ethynyl groups were attached to the porphyrin core as a linker capable for inducing an extensive conjugation through π−π orbital interactions favourable to charge delocalization between the macrocycle and (hetero)aromatic units with the goal of achieving subsequent tailored electronic properties of porphyrin-based materials. The ethynyl bridges were grafted in the *β* [18] but mainly in the *meso* positions of the porphyrine core [19,20,21,22]. Substituents attached at the level of the peripheral porphyrin ring often caused minor structural changes without modifying the planar geometry of the porphyrin core, which instead could be affected by bulky aromatic substituents on *meso* or *β* positions and also by introducing substituents at the inner pyrrole nitrogen atom. It was documented that under acidic conditions, such as trifluoroacetic acid (TFA) [23] or methanesulfonic acid (MSA) [24] in homogeneous organic solvent solutions, the porphyrin core (H_2_Por) may undergo protonation at one or both nitrogen atoms of the pyrrole units, thus resulting in either monocationic (H_3_Por^+^) or dicationic (H_4_Por^2+^) species [25,26,27] displaying perturbed photophysical properties compared to their neutral parent compounds, including altered electronic absorption spectra, substantially lowered quantum yield of triplet-state formation, and increased Stokes-shifts in fluorescence spectra [12,13]. The core protonation of porphyrin contributes to an out-of-plane tilting of the individual pyrrole rings, as the four hydrogen atoms do not fit into the central cavity of the macrocycle and thus generate the so-called hyperporphyrin spectra, which are characterized by a broadened and/or split Soret band along with an intense new red-shifted absorption replacing the typical Q band situated in the visible spectral region. Electron-donating functional groups situated in the *meso* positions of the porphyrin core considerably increased the protonation tendency at the level of pyrrole rings and the chemical stability of the resulted cationic species [28]. Insights on the protonation behaviour of porphyrin macrocycle, molecular geometries, electronic structures, vibrational spectra, etc., were also brought by theoretical molecular modelling (DFT). It was found that the angles between the planes of aryl substituents and the porphyrin core substantially decrease upon protonation of the inner nitrogen atoms, leading to enhanced resonance interactions between π-systems of porphyrin and aryl substituents. The second protonation is particularly facilitated in case of tertraphenylporphyrin (TPP) by the large out-of-plane flexibility of the diprotonated species as unraveled by ab initio molecular dynamics [29]. The results of molecular modeling studies using the density functional theory described the ground-state structure of porphyrin diacid (H_4_Por^2+^) as a stable saddle-distorted porphyrin ring (*D*_2d_ symmetry) with the four pyrrole rings tilted up and down alternately with respect to the porphyrin mean plane and time-dependent DFT (TD-DFT) computations’ predicted excitation energies and oscillator strengths, at least for the first and second excited states, which are in good agreement with the experimental electronic absorption spectrum [30].

Encompassing our study on the topic of synthesis, metal complexation and optical properties of *meso*-phenothiazinyl-phenyl porphyrin (MPP) dyes previously reported by our research group [31,32], in this work we introduce a series of additional new MPP derivatives with extended-conjugation π-electron systems brought by peripheral/bridging ethyne auxochrome units capable of inducing favorable steric orientation of the aromatic rings. Their synthesis based on direct one-pot Adler–Longo mixt. condensation reaction of pyrrole with ethyne functionalized (hetero)aryl-carbaldehydes was considered as a more advantageous alternative to Sonogashira cross coupling reaction of an alkyne with halogen substituted MPP due to the low solubility of the porphyrin substrate. The variation of optical UV-Vis absorption/emission properties in the state of free bases and protonated species are discussed. 

## 2. Results and Discussions

### 2.1. Chemical Synthesis

A series of new MPP comprising peripheral ethynyl units (ethynyl-MPP) was successfully prepared by one-pot Adler–Longo mixt. mixed condensation reaction of pyrrole with ethynyl functionalized (hetero)aryl-carbaldehydes (7-ethynyl-10-methyl-10*H*-phenothiazin-3-carbaldehyde [33] or 4-ethynyl-benzaldehyde) and (hetero)aryl-carbaldehydes (7-bromo-10-methyl-10*H*-phenothiazin-3-carbaldehyde [34] or 4-bromo-benzaldehyde); mixtures of A_3_B- (**2a**, **2b**, **2c**, **2d**) and *trans*-A_2_B_2_-type (**3a**, **3b**, **3c**, **3d**) ethynyl-MPP dyes were thus obtained in a 2:1 molar ratio (Scheme 1). *Trans* disubstituted isomers were selectively obtained as major rection products pointing towards a steric control induced by the folded structure of phenothiazine-carbaldehyde [31].

Taking advantage of the efficacy of the porphyrin synthesis by direct one-pot Adler–Longo mixt. mixed condensation reaction, a convergent synthetic strategy was designed for the preparation of new porphyrin derivatives with extended conjugated π-electron systems pending to the porphyrin core. Thus, ethynylene linked (hetero)aryl carbaldehydes **6** or **7** prepared by Sonogashira cross-coupling reaction (Scheme 2) were further used as starting materials for porphyrin synthesis. TPP-ethynylene-phenothiazine conjugates (**4a**, **5a**) were obtained using phenothiazine-ethynylene-benzaldehyde conjugate **6** as a starting carbaldehyde and MPP-ethynylene-dibromobenzene conjugates (**4b**, **5b**) were obtained starting with dibromobenzene-ethynylene-phenothiazine carbaldehyde conjugate **7** respectively (Scheme 3). The purification of the target porphyrins from the complex reaction mixture was achieved after repeated column chromatography separations (A_3_B- and A_2_B_2_-type porphyrins were eluted as the second and the third fraction, respectively).

### 2.2. UV-Vis Absorption and Emission Properties of Ethynyl Functionalized MPP Derivatives 2A–L

The UV-Vis absorption spectra of the dark purple ethynyl-MPP **2a–d**, **3a–d** displayed a high intensity absorption band with maxima in the UV spectral range (250–253 nm) assigned to the presence of the phenothiazine chromophore followed by a strong near-UV Soret band (419–424 nm) accompanied by four low intensity *etio*-type Q bands situated in the visible spectral region (514–659 nm) representative for the porphyrin chromophore, as exemplified in Figure 1. A close inspection of the recorded absorption maxima summarized in Table 1 disclosed a minor bathochromic shift of the Soret band (up to 4 nm) upon peripheral functionalization of parent MPP (λ_abs_ 420 nm [31]) with ethynyl units, best represented by *trans*-A_2_B_2_-type ethynyl-MPP **3a** with peripheral functionalization of the phenyl units with electron donor trimethylsilyl-ethynyl substituents (λ_abs_ 424 nm). Very similar absorption spectra were recorded upon extending the TPP/MPP chromophore system via ethynylene bridges with electron-rich phenothiazine (**4a**, **5a**) or dibromophenylene auxochrome units (**4b**, **5b**). These spectral features indicate that the UV-Vis absorption spectra were produced mainly by the porphyrin chromophore system, the substituents participating with feeble electronic/steric auxochrome effects. Higher molar absorptivity values were exhibited by A_3_B- as compared to A_2_B_2_-type porphyrins due to their less symmetrical structure. 

Upon excitation with the Soret band maxima in dichloromethane solution, each MPP derivative displayed red-orange daylight fluorescence (Figure 2) characterized by large Stokes shift (8410–8691 cm^−1^) and noticeable quantum yields (***Φ*_F_** 0.11–0.25 against TPP standard) (Table 1), thus identifying them as good candidates for red-emitting materials.

The phenothiazine fluorophore displayed greenish daylight fluorescence characterized by large Stokes shifts but low fluorescence quantum yields (e.g., phenothiazine-carbaldehyde acetals λ_em_ 506 nm, ***Φ***_fl_ < 0.01) [35], while *meso*-tetraphenothiazinyl-porphyrin exhibited electronic spectral characteristics typical to the porphyrin fluorophore with a red shift of the emission maxima (λ_em_ = 669 nm, ***Φ***_fl_ = 0.05 [34]) comparative to TPP (λ_em_ 652 nm ***Φ***_fl_ = 0.11). Functionalization of the *meso* positions of the porphyrin with ethynyl groups resulted in a significant bathochromic shift of the fluorescence peak maxima and higher fluorescence quantum yields (e.g., 5,15- bis(arylethynyl)-10,20-diphenyl-porphyrine λ_em_ = 695 nm, ***Φ***_fl_ = 0.20 [23]). In the case of the newly reported compounds, by joining the TPP system via ethynylene bridges with phenothiazine units, the position of the emission maxima appeared comparable to TPP but the fluorescence quantum yield’s value became significantly higher (e.g., for **5a** λ_em_ = 656 nm, ***Φ***_fl_ > 0.25 Table 1), situated among the highest values observed for porphyrin derivatives.

### 2.3. Optical Properties of the Protonated Ethynyl-MPP Derivatives 

The potential changes of the photophysical properties of the novel ethynyl-MPP derivatives in the presence of strong acids were investigated. Protonation of ethynyl-MPP free bases with trifluoroacetic acid (TFA) were carried out in DCM solution and monitored by UV-vis absorption/emission spectroscopy. Upon stepwise addition of increasing amounts of TFA, the modifications that occurred in the UV-vis absorption spectra are consistent with the spectral features exhibited by hyperporphyrins [25], suggesting the formation of a dication caused by attaching two protons to the inner nitrogen atoms of the macrocycle (H_2_MPP^2+^); thus, the Soret band appeared intensified and red-shifted with 19 nm, while the multiple Q bands collapsed and a broad Q band was displayed above 680 nm. In each case, a distinct color change from brown to green was observable with the naked eye. As exemplified in Figure 3, showing the modifications recorded in the UV-Vis spectra of the free-bases A_3_B-type **2a** with peripheral ethynyl functionalization (Figure 3a) and A_2_B_2_-type TPP-ethynyl-phenothazine conjugate **5a** (Figure 3b), respectively, upon protonation with increasing amounts of TFA (from 10 μL up to 100 μL), the Soret band appeared more intense and red-shifted with 19 nm with the occurrence of an isosbestic point, while the position of the red hyperporphyrin absorbance situated above 680 nm appeared to be influenced by the degree of protonation. We assumed that the addition of 10 μL TFA ensured the protonation at the porphyrin core and facilitated the formation of dications **H_2_2a^2^**^+^ (λ_max_ = 682 nm) and **H_2_5a^2+^** (λ_max_ = 688 nm), respectively, characterized by a D–A charge transfer between the donor phenothiazine peripheral unit(s) and the acceptor protonated porphyrin core. Further addition of more than 30μL TFA produced a noticeable hypsochromic shift of the red absorption band and this could be the effect of the protonation at the nitrogen atom in the phenothiazine unit(s), which suppressed the electron donor effect of the phenothiazine moiety in **H_3_2a^3+^** (λ_max_ = 679 nm) and **H_4_5a^4+^** (λ_max_ = 649 nm), respectively. This assumption designates phenothiazine as a weaker base in comparison to porphyrin.

In Figure 4 are shown the hyperporphyrin spectra of **H_2_TPP^2+^** and **H_2_MPP^2^**^+^ obtained from TPP and A_3_B-type porphyrins **2a**, **2c**, **4a**, **4c** respectively, upon protonation with TFA in DCM solution. A comparison of the position of the Soret and Q_4_ absorption bands displayed in the spectra of the free bases **2a**, **2c**, **4a**, **4c** (Table 1) with those of the corresponding protonated species (Figure 4) indicate bathochromic shifts of the Soret band (19–26 nm) and significant red shifts of the corresponding Q band (151–176 nm) as a consequence of the core protonation of the porphyrin, in agreement with literature data mentioning these distinctive electronic characteristics upon protonation of TPP and porphyrin functionalized in *meso* positions with ethynyl linkages [25]. Almost superimposed Soret bands are observable for similar **H_2_2a^2+^** and **H_2_2c^2+^** (448 nm) but the long wave maxima displayed a 32 nm bathochromic shift for **H_2_2a^2+^** containing the electron donor trimethylsilyl substituents. A bathochromic shift of the Soret band was observed for the protonated TPP-ethynyl-phenothiazine conjugate (**H_2_4a^2+^**) as compared to **H_2_TPP**^2+^, indicative of an elongation of the conjugated π-electron system by the electron donor phenothiazine unit. The extension of the conjugated π-electron system at the level of the pending phenothiazine substituent did not produce notable modifications in the absorption spectrum of **H_2_4b^2+^** as compared to **H_2_TPP^2+^**.

The fluorescence emission spectra of **H_2_MPP^2+^** presented red-shifted and broadened emission bands with reduced intensity (Figure 5a), except for the case of **2c** (Figure 5b), which provided clear evidence of efficient energy transfer processes by peripheral substitution with ethynyl groups. 

Based on the premise that electronic spectra of porphyrin compounds are routinely interpreted with the four-orbital model of Gouterman [36], assuming that the Soret (B-) and Q- bands are generated by one-electron excitations from HOMO/HOMO-1 (nearly degenerate) to the LUMO (strictly degenerate orbitals), an insight of the electronic features of the frontier molecular orbitals of the symmetrical TPP-ethynyl-phenothiazine conjugate **5a** as free base and corresponding mono- and diprotonated species was achieved by the DFT method. The results of our B3LYP-DFT calculations pointed out the ground-state of the most stable conformer of **H_2_5a^2+^** containing a saddle-distorted porphyrin ring (*D*_2d_ symmetry) with the four pyrrole rings tilted up and down alternately with respect to the porphyrin mean plane, similar to **H_2_TPP^2+^** [24]. As it can be seen in Figure 6, presenting the plots of the FMO of **H_2_5a^2+^**, the spin density in HOMO/HOMO-1 (separated by 0.0043 eV) appeared delocalized mainly on the electron donor phenothiazine units, while the LUMO/LUMO+1 (separated by only 0.0013 eV) are mainly centered on the porphyrin core. 

## 3. Materials and Methods 

The reagents and solvents were purchased from commercial sources, and pyrrole was redistilled before use. All reactions were followed by thin layer chromatography (TLC) analysis using Merck pre-coated silica gel 60 F_254_ aluminum sheets. Column chromatography was performed on silica (60 Å, particle size 0.063‒0.2 mm). 

One-dimensional, 2D-COSY, 2D-HMQC and 2D-HMBC NMR spectra were recorded on Bruker Avance instruments (400 and 600 MHz) using deuterated solvents (CDCl_3_ and DMSO) at room temperature (NMR spectra of compounds **2a**, **3a**, **2c**, **3c**, **4a** are shown in Appendix A), chemical shifts are quoted in parts per million (ppm) relative to tetramethylsilane (TMS) and *J* values are given in Hz. Mass spectra were measured on a HRMS spectrometer LTQ-Orbitrap XL-Thermo-Scientific using APCI or ESI ionization techniques (HRMS spectra of compounds **3a**, **2c**, **3c**, **5a** are shown in Appendix A). UV-Vis spectra were recorded in dichloromethane with a Perkin Elmer Lambda 35 spectrophotometer, fluorescence spectra measured in dichloromethane using a Perkin Elmer 55 PL spectrophotometer. The fluorescence quantum yield was calculated using tetraphenylporphyrin (TPP) standard (***Φ*** = 0.13 CH_2_Cl_2_ solution). 

DFT Calculation Geometries were fully optimized using the density functional theory (DFT). The functional and basis set used in the DFT calculations were the Becke’s three-parameter hybrid functional combined with the Lee–Yang–Parr correlation functional (B3LYP) and the 6-31G(d,p) basis set, respectively. Equilibrium geometries were verified via frequency calculations, where no imaginary frequency was found. All the calculations were carried out using the Gaussian 09 program suite.

7-Ethynyl-10-methyl-10H-phenothiazin-3-carbaldehyde **6** was prepared according to previously reported procedures [33].

7-Bromo-10-methyl-10H-phenothiazin-3-carbaldehyde **7** was prepared according to previously reported procedures [34].

### General Procedure for the Synthesis of 2A–L

Propanoic acid and acetic anhydride were stirred and heated at 110 °C for 1 h. After cooling at room temperature, aryl-aldehyde (*2eq*), 10-methyl-10*H*-phenothiazinyl-carbaldehyde derivative (*2eq*) and pyrrole (*4eq*) were added and the mixture was heated at 110 °C for 4 h. After completion, the obtained purple crystals were collected by filtration and washed with methanol to remove the traces of propionic acid. In the case when the product did not precipitate, the solvent was removed by reduced pressure distillation and the residue was washed with methanol. The further purification and separation by column chromatography on silica using dichloromethane/petrol ether (1:2) gave the corresponding A_3_B- and A_2_B_2_-type compounds. Due to the fact that the retention factors of the A_3_B-type compounds (Rf = 0.4) and A_2_B_2_-type (Rf = 0.3) are pretty close, 3–4 successive column chromatography separations are required in order to achieve high purity products.

*5,10,15-tri(trimethylsilylethynyl-phenyl)-20-(7-bromo-10-methyl-10H-phenothiazin-3-yl)-21,23H-porphyrin* (**2a**). Purple powder, yield 12% (0.2 g).^1^H-NMR (600 MHz, CDCl_3_) δppm −2.72 (s, 2H, NH), 0.46 (s, 27H), 3.54 (s, 3H), 6.78 (d, 1H, *J* = 8.7 Hz), 7.12 (d, 1H, *J* = 8.1 Hz), 7.37 (dd, 1H, *J* = 8.6 Hz, *J* = 2.1 Hz), 7.40 (s, 1H), 7.94 (d, 6H, *J* = 7.1 Hz), 7.98 (dd, 1H, *J* = 8.0 Hz, *J* = 1.8 Hz), 8.02 (s, 1H), 8.22 (d, 6H, *J* = 6.7 Hz), 8.88 (d, 6H, *J* = 4.5 Hz), 8.95 (d, *J* = 4.3 Hz); ^13^C-NMR (150 MHz, CDCl_3_) δppm 0.1 (9C), 35.6 (C), 95.7 (C_q_), 105.0 (C_q_), 112.5 (C_q_), 114.9 (C), 115.1 (C_q_), 115.2 (3C), 115.4 (C_q_), 119.3 (C_q_), 119.5 (C_q_), 119.6 (C), 121.3 (C_q_), 122.7 (C), 122.8 (C_q_), 124.8 (2C_q_), 125.5 (C_q_), 129.4 (4C), 129.7 (2C_q_), 130.3 (4C), 130.3 (2C_q_), 130.4 (6C), 132.9 (C_q_), 134.0 (C_q_), 134.4 (6C), 136.6 (C_q_), 142.3 (C_q_), 142.3 (4C_q_), 144.5 (2C_q_), 145.0 (2C_q_), 145.2 (2C_q_). HRMS-APCI Calcd for: C_66_H_59_BrN_5_SSi_3_ [M + H]^+^ 1116.29769, Found: 1116.29553.

*5,15-di(trimethylsilylethynyl-phenyl)-10,20-di-(7-bromo-10-methyl-10H-phenothiazin-3-yl)-21,23H-porphyrin* (**3a**). Purple powder, yield 12% (0.2 g).^1^H-NMR (600 MHz, CDCl_3_) δppm −2.70 (s, 2H, NH), 0.45 (s, 18H), 3.49 (s, 6H), 6.76 (d, 2H, *J* = 8.7 Hz), 7.03–7.07 (m, 2H), 7.34 (d, 2H, *J* = 2.1Hz), 7.39 (s, 2H), 7.93–7.95 (m, 4H), 7.95 (d, 2H, *J* = 7.7 Hz), 8.02 (s, 2H), 8.21 (d, 4H, *J* = 6.5Hz), 8.87–8.94 (m, 8H); ^13^C-NMR (150 MHz, CDCl_3_) δppm 0.1 (6C), 35.6 (2C), 95.6 (2C_q_), 105.0 (2C_q_), 112.4 (2C), 115.0 (2C), 115.4 (4C), 119.1 (C_q_), 119.2 (2C_q_), 119.4 (2C_q_), 119.5 (C_q_), 121.3 (C_q_), 122.7 (C_q_), 122.7 (2C), 125.5 (2C), 129.7 (4C), 130.3 (4C), 130.4 (4C), 132.9 (2C_q_), 132.9 (C_q_), 134.0 (2C_q_), 134.0 (C_q_), 134.4 (4C), 136.6 (2C_q_), 136.6 (C_q_), 142.3 (2C_q_), 142.4 (C_q_), 144.9 (4C_q_), 145.1 (2C_q_), 145.1 (2C_q_). HRMS-APCI Calcd for: C_68_H_55_Br_2_N_6_S_2_Si_2_ [M + H]^+^ 1235.18090, Found: 1235.17603.

*5,10,15-tri(4-bromophenyl)-20-(7-trimethylsilylethynyl-10-methyl-10H-phenothiazin-3-yl)-21,23H-porphyrin* (**2b**). Purple powder, yield 18% (0.2 g).^1^H-NMR (600 MHz, CDCl_3_) δppm −2.81 (s, 2H, NH), 0.29 (s, 9H), 3.62 (s, 3H), 6.86 (d, 1H, *J* = 8.5 Hz), 7.21 (d, 1H, *J* = 7.0 Hz), 7.38–7.43 (m, 2H), 7.91 (d, 6H, *J* = 7.2 Hz), 7.98–7.99 (m, 2H), 8.08 (d, 6H, *J* = 6.6 Hz), 8.85 (s, 6H), 8.93 (d, *J* = 2.7 Hz); ^13^C-NMR (150 MHz, CDCl_3_) δppm 0.05 (3C), 35.7 (C), 90.0 (C_q_), 94.2 (C_q_), 104.3 (C_q_), 112.5 (C), 113.9 (C_q_), 115.1 (C_q_), 115.5 (C_q_), 118.7 (C_q_), 118.8 (C), 119.4 (C_q_), 121.4 (C_q_), 121.5 (C_q_), 122.6 (2C), 123.0 (C_q_), 125.3 (2C), 125.4 (C_q_), 128.2 (4C), 129.0 (4C), 129.9 (6C), 130.3 (C_q_), 130.6 (C_q_), 131.6 (C_q_), 133.9 (C_q_), 134.0 (C_q_), 135.8 (6C), 136.5 (C_q_), 136.5 (C_q_), 137.8 (C_q_), 140.9 (4C_q_), 145.2 (C_q_), 145.8 (C_q_). HRMS-APCI Calcd for: C_56_H_41_Br_3_N_5_SSi [M + H]^+^ 1082.03761, Found: 1082.03333.

*5,15-di(4-bromophenyl)-10,20-di(7-trimethylsilylethynyl-10-methyl-10H-phenothiazin-3-yl)-21,23H-porphyrin* (**3b**). Purple powder, yield 16% (0.2 g).^1^H-NMR (400 MHz, CDCl_3_) δppm −2.78 (s, 2H, NH), 0.31 (s, 18H), 3.55 (s, 6H), 6.80 (d, 2H, *J* = 9.2 Hz), 7.21 (d, 2H, *J* = 7.9 Hz), 7.36–7.38 (m, 4H), 7.89 (d, 4H, *J* = 7.4 Hz), 7.95–7.96 (m, 2H), 7.99 (s, 2H), 8.07 (d, 4H, *J* = 7.3 Hz), 8.83-8.93 (m, 8H); ^13^C-NMR (100 MHz, CDCl_3_) δppm 0.05 (6C), 35.6 (2C), 94.2 (2C_q_), 104.2 (2C_q_), 112.5 (2C), 115.1 (2C), 115.2 (2C), 118.6 (2C_q_), 119.3 (2C_q_), 122.5 (2C), 125.3 (2C), 125.4 (2C), 128.2 (4C), 129.0 (4C), 129.7 (4C_q_), 129.9 (4C), 130.4 (4C_q_), 130.6 (2C_q_), 130.9 (2C_q_), 131.6 (2C_q_), 135.8 (4C), 136.5 (2C_q_), 137.9 (2C_q_), 140.9 (2C_q_), 145.0 (2C_q_), 145.1 (2C_q_).HRMS-APCI Calcd for: C_68_H_55_Br_2_N_6_S_2_Si_2_ [M+H]^+^ 1233.18294, Found: 1233.17751.

*5,10,15-tri(ethynyl-phenyl)-20-(7-bromo-10-methyl-10H-phenothiazin-3-yl)-21,23H-porphyrin* (**2c**). Purple powder, yield 14% (0.2 g).^1^H-NMR (600 MHz, CDCl_3_) δppm −2.79 (s, 2H, NH), 3.35 (s, 3H), 3.62 (s, 3H), 6.86 (d, 1H, *J* = 8.5 Hz), 7.18 (d, 1H, *J* = 8.6 Hz), 7.39–7.41 (m, 2H), 7.92 (d, 6H, *J* = 6.4 Hz), 8.00–8.01 (m, 2H), 8.20 (d, 6H, *J* = 8.8 Hz), 8.86 (d, 6H, *J* = 5.2 Hz), 8.93 (d, 2H, *J* = 4.0 Hz); ^13^C-NMR (150 MHz, CDCl_3_) δppm 35.7 (C), 78.4 (3C), 83.6 (3C_q_), 112.5 (C), 115.1 (2C_q_), 115.5 (C), 119.3 (C_q_), 119.3 (C_q_), 119.4 (C), 119.5 (C), 121.3 (2C_q_), 121.7 (2C), 121.8 (2C_q_), 125.4 (C_q_), 128.2 (C_q_), 129.0 (C_q_), 129.7 (C), 130.3 (C), 130.5 (8C), 132.9 (C), 134.0 (C), 134.4 (8C), 136.5 (C_q_), 142.5 (2C_q_), 142.6 (2C_q_), 142.6 (4C_q_), 145.0 (2C_q_), 145.2 (2C_q_). HRMS-APCI Calcd for: C_57_H_35_BrN_5_S [M + H]^+^900.17911, Found: 900.17433.

*5,15-di(ethynyl-phenyl)-10,20-di-(7-bromo-10-methyl-10H-phenothiazin-3-yl)-21,23H-porphyrin* (**3c**). Purple powder, yield 15% (0.2 g).^1^H-NMR (600 MHz, CDCl_3_) δppm −2.75 (s, 2H, NH), 3.35 (s, 2H), 3.55 (s, 6H), 6.80 (d, 2H, *J* = 8.7 Hz), 7.10–7.12 (m, 2H), 7.37 (d, 4H, *J* = 8.1 Hz), 7.91 (d, 4H, *J* = 6.7 Hz), 7.96 (d, 2H, *J* = 7.8 Hz), 8.00 (s, 2H), 8.19 (d, 4H, *J* = 6.7 Hz), 8.85–8.92 (m, 8H); ^13^C-NMR (150 MHz, CDCl_3_) δppm 35.6 (2C), 78.4 (2C), 83.6 (2C_q_), 112.5 (2C), 115.0 (2C), 115.4 (2C), 119.2 (2C_q_), 119.3 (2C_q_), 121.3 (2C_q_), 121.3 (2C_q_), 121.7 (2C), 125.4 (2C_q_), 129.7 (4C), 130.3 (4C), 130.5 (6C), 132.9 (2C_q_), 132.9 (C_q_), 134.0 (2C_q_), 134.0 (C_q_), 134.4 (6C), 136.6 (2C_q_), 136.6 (C_q_), 142.6 (2C_q_), 145.0 (3C_q_), 145.1 (C_q_), 145.1 (3C_q_). HRMS-APCI Calcd for: C_62_H_39_Br_2_N_6_S_2_ [M + H]^+^1091.10184, Found: 1091.09713.

*5,10,15-tri(4-bromophenyl)-20-(7-ethynyl-10-methyl-10H-phenothiazin-3-yl)-21,23H-porphyrin* (**2d**). Purple powder, yield 14% (0.1 g).^1^H-NMR (600 MHz, CDCl_3_) δppm −2.79 (s, 2H, NH), 3.57 (s, 1H), 3.65 (s, 3H), 6.98 (dd, 1H, *J* = 8.6 Hz, *J* = 3.3 Hz),7.20 (d, 1H, *J* = 7.0 Hz), 7.37–7.38 (m, 2H), 7.83 (s, 1H), 7.89–7.91 (m, 6H), 7.99 (dd, 2H, *J* = 7.9 Hz, *J* = 2.0 Hz), 8.01 (d, 1H, *J* = 1.8 Hz), 8.07 (d, 4H, *J* = 7.0 Hz), 8.85–8.93 (m, 8H); ^13^C-NMR (150 MHz, CDCl_3_) δppm 36.0 (C), 81.0 (C), 84.8 (C_q_), 111.7 (C_q_), 112.1 (C_q_), 116.0 (C), 116.1 (C), 118.8 (C_q_), 118.9 (2C_q_), 119.2 (C_q_), 121.3 (C_q_), 121.4 (C_q_), 122.5 (2C_q_), 122.6 (C), 123.0 (C), 125.3 (C), 127.5 (C), 128.2 (2C), 128.2 (4C), 129.0 (4C), 129.7 (C_q_), 131.7 (6C), 131.9 (C_q_), 134.0 (C_q_), 134.0 (C_q_), 135.8 (4C), 139.1 (C_q_), 139.8 (C_q_), 140.9 (C_q_), 145.1 (2C_q_), 145.3 (C_q_), 145.7 (C_q_), 146.0 (C_q_), 148.8 (2C_q_). HRMS-APCI Calcd for: C_53_H_33_Br_3_N_5_S [M+H]^+^ 1011.63168, Found: 1011.99146.

*5,15-di(4-bromophenyl)-10,20-di(7-ethynyl-10-methyl-10H-phenothiazin-3-yl)-21,23H-porphyrin* (**3d**). Purple powder, yield 14% (0.1 g) ^1^H-NMR (400 MHz, CDCl_3_) δppm 2.79 (s, 2H, NH), 2.38 (s, 2H), 3.66 (s, 6H), 7.14–7.21 (m, 4H), 7.38 (d, 2H, *J* = 6.2 Hz), 7.83 (s, 2H), 7.90 (d, 4H, *J* = 6.4 Hz), 7.98–8.01 (m, 4H), 8.08 (d, 4H, *J* = 7.8 Hz), 8.85–8.94 (m, 8H); ^13^C-NMR (100 MHz, CDCl_3_) δppm 36.0 (2C), 112.5 (C_q_), 112.9 (3C), 113.7 (3C), 115.5 (C_q_), 118.8 (C_q_), 118.9 (2C_q_), 119.2 (C_q_), 121.3 (C_q_), 121.4 (C_q_), 122.5 (2C_q_), 122.6 (2C), 123.0 (2C), 125.3 (2C), 127.7 (2C), 128.2 (4C), 128.8 (C_q_), 129.0 (4C), 129.7 (C_q_), 130.0 (4C), 131.9 (C_q_), 132.9 (2C_q_), 132.9 (2C_q_), 134.0 (C_q_), 134.0 (C_q_), 135.8 (4C), 137.1 (C_q_), 137.9 (C_q_), 140.9 (C_q_), 140.9 (2C_q_), 144.2 (2C_q_), 144.2 (2C_q_), 149.8 (2C_q_). HRMS-APCI Calcd for: C_62_H_39_Br_2_N_6_S_2_ [M + H]^+^ 1091.10184, Found: 1091.12134.

*5,10,15-triphenyl-20-((10-methyl-10H-phenothiazin-3-yl)ethynylphenyl-4-yl)-21,23H-porphyrin* (**4a**). Purple powder, yield 16% (0.06 g).^1^H-NMR (600 MHz, CDCl_3_) δppm −2.72 (s, 2H, NH), 3.45 (s, 3H), 6.85 (d, 1H, *J* = 8.2 Hz), 6.87 (d, 1H, *J* = 8.1 Hz), 7.01 (t, 1H, *J* = 7.5 Hz),7.20–7.24 (m, 2H), 7.48 (s, 1H), 7.50 (dd, 1H, *J* = 8.1 Hz, *J* = 1.8 Hz),7.77-7.82 (m, 9H), 7.92 (d, 2H, *J* = 7.9 Hz), 8.23 (d, 2H, *J* = 7.9 Hz), 8.25 (d, 6H, *J* = 6.4 Hz), 8.90 (d, 8H, *J* = 8.1 Hz); ^13^C-NMR (150 MHz, CDCl_3_) δppm 35.4 (C), 89.3 (C_q_), 90.1 (C_q_), 113.9 (2C), 114.3 (2C), 117.2 (2C_q_), 119.3 (C_q_), 119.4 (C_q_), 120.2 (2C_q_), 120.3 (C_q_), 120.4 (C_q_), 122.9 (4C), 122.9 (2C_q_), 122.9 (2C_q_), 123.6 (2C_q_), 126.7 (6C), 127.2 (2C), 127.6 (C), 127.7 (C), 129.8 (C), 130.3 (C), 131.2 (C), 134.5 (9C), 134.6 (4C), 141.9 (C_q_), 142.0 (C_q_), 142.1 (2C_q_), 145.2 (2C_q_), 146.0 (2C_q_). HRMS-APCI Calcd for: C_59_H_40_N_5_S [M + H]^+^ 850.29989, Found: 850.29675.

*5,15-diphenyl-10,20-di((10-methyl-10H-phenothiazin-3-yl)ethynylphenyl-4-yl)-21,23H-porphyrin* (**5a**). Purple powder, yield 14% (0.06 g).^1^H-NMR (600 MHz, CDCl_3_) δppm −2.75 (s, 2H, NH), 3.46 (s, 6H), 6.86–6.89 (m, 4H), 7.01 (t, 2H, *J* = 7.4 Hz), 7.20–7.25 (m, 4H), 7.48 (s, 2H), 7.50 (d, 2H, *J* = 8.1 Hz), 7.77–7.81 (m, 6H), 7.91-7.93 (m, 4H), 8.21–8.24 (m, 8H), 8.87-8.90 (m, 8H), ^13^C-NMR (150 MHz, CDCl_3_) δppm 36.0 (2C), 90.0 (2C_q_), 96.1 (2C_q_), 113.9 (C_q_), 114.0 (2C), 114.1 (C_q_), 114.5 (C_q_), 114.6 (2C), 116.2 (2C), 116.3 (C_q_), 117.9 (C_q_), 118.4 (C_q_), 122.1 (C_q_), 122.9 (C_q_), 123.5 (2C), 123.5 (2C), 128.0 (4C), 128.0 (2C), 128.6 (C_q_), 128.7 (C_q_), 129.6 (4C), 129.7 (10C), 130.5 (C_q_), 130.5 (2C), 131.8 (C_q_), 132.3 (C_q_), 133.7 (8C), 135.8 (C_q_), 143.2 (C_q_), 144.5 (C_q_), 150.0 (2C_q_), 150.3 (2C_q_), 165.3 (8C_q_). HRMS-APCI Calcd for: C_74_H_49_N_6_S_2_ [M + H]^+^ 1085.34546 Found: 1085.64795.

*5,10,15-tri(4-bromophenyl)-20-(7-((3,5-dibromophenyl)ethynyl)-10-methyl-10H-phenothiazin-3-yl)-21,23H-porphyrin* (**4b**). Purple powder, yield 18% (0.3 g).^1^H-NMR (600 MHz, CDCl_3_) δppm −2.81 (s, 2H, NH), 3.66 (s, 3H), 6.86 (d, 1H, *J* = 8.7 Hz), 6.96 (d, 1H, *J* = 8.5 Hz), 7.16–7.19 (m, 1H), 7.40 (dd, 1H, *J* = 7.59 Hz, *J* = 1.9 Hz),7.45–7.46 (m, 1H), 7.63 (d, 1H, *J* = 1.7 Hz), 7.64 (t, 1H, *J* = 1.7 Hz), 7.91 (d, 6H, *J* = 7.9 Hz), 7.99–8.00 (m, 2H), 8.08 (d, 6H, *J* = 7.8 Hz), 8.86–8.95 (m, 8H), ^13^C-NMR (150 MHz, CDCl_3_) δppm 35.7 (C), 86.6 (C_q_), 91.4 (C_q_), 112.5 (C_q_), 112.6 (C), 114.2 (C), 115.1 (C_q_), 115.5 (C_q_), 116.5 (C_q_), 118.7 (C_q_), 118.9 (2C), 119.4 (C_q_), 121.3 (C_q_), 121.4 (C_q_), 122.6 (4C), 122.6 (2C), 123.4 (C_q_), 125.4 (C_q_), 126.8 (C_q_), 129.7 (C_q_), 129.9 (6C), 130.3 (C), 130.3 (C_q_), 131.5 (C_q_), 132.8 (4C), 132.9 (2C), 133.7 (C_q_), 133.9 (C_q_), 134.0 (C_q_), 135.8 (6C), 136.7 (C_q_), 140.8 (C_q_), 140.9 (4C_q_), 144.8 (C_q_), 144.9 (C_q_), 145.2 (C_q_), 146.3 (C_q_). HRMS-APCI Calcd for: C_59_H_35_Br_5_N_5_S [M + H]^+^ 1245.84687, Found: 1245.84509.

*5,15-di(4-bromophenyl)-10,20-di(7-((3,5-dibromophenyl)ethynyl)-10-methyl-10H-phenothiazin-3-yl)-21,23H-porphyrin* (**5b**). Purple powder, yield 16% (0.3 g), ^1^H-NMR (600 MHz, CDCl_3_) δppm −2.79 (s, 2H, NH), 3.60 (s, 3H) 3.65 (s, 3H), 6.85 (dd, 1H, *J* = 8.6 Hz, *J* = 1.3Hz), 6.95 (dd, 1H, *J* = 8.3 Hz, *J* = 1.6 Hz), 7.16–7.18 (m, 3H), 7.39–7.41 (m, 3H), 7.44–7.45 (m, 2H), 7.62 (s, 2H), 7.64 (s, 2H), 7.91 (d, 4H, *J* = 7.3 Hz), 8.00 (s, 4H), 8.09 (d, 4H, *J* = 6.7 Hz), 8.84–8.93 (m, 8H); ^13^C-NMR (150 MHz, CDCl_3_) δppm 35.8 (2C), 86.6 (2C_q_), 91.4 (2C_q_), 112.6 (C_q_), 114.2 (2C), 115.1 (C_q_), 115.5 (C_q_), 116.5 (C_q_), 118.6 (2C_q_), 118.7 (C_q_), 119.3 (C_q_), 119.3 (C_q_), 121.3 (C_q_), 121.3 (C_q_), 122.5 (2C), 122.6 (4C), 123.4 (2C_q_), 125.4 (C_q_), 126.8 (2C), 129.7 (2C_q_), 129.9 (4C), 130.3 (2C_q_), 130.3 (2C_q_), 131.5 (2C), 132.8 (8C), 132.9 (2C), 133.0 (2C_q_), 133.7 (2C), 134.0 (C_q_), 134.0 (C_q_), 135.8 (4C), 136.5 (C_q_), 136.8 (C_q_), 140.9 (2C), 144.8 (2C_q_), 145.0 (2C_q_), 145.2 (2C_q_), 146.3 (2C_q_). HRMS-APCI Calcd for: C_74_H_43_Br_6_N_6_S_2_ [M + H]^+^1558.80240, Found: 1558.80615.

## 4. Conclusions

This one-pot mixt. condensation of pyrrole with ethynyl functionalized (hetero)aryl-carbaldehydes is recommended as an advantageous synthetic strategy for the preparation of ethyne functionalized MPP in 12–16% yields. 

Feeble auxochromic effects were displayed in the UV-Vis absorption/emission spectra of the new ethynyl-MPP, which were shaped mainly by the porphyrin chromophore system characterized by intense Soret bands situated at 419–424 nm, four low intensity *etio* type Q bands situated in the range of 514–651 nm, and intensified fluorescence emission in the range of 651–659 nm with 11–25% quantum yields. These fluorescence emission properties open perspectives for staining biological tissues in the first NIR therapeutic window. 

Electron-donor phenothiazine units situated in the *meso* positions of the porphyrin core favoured the protonation at the level of pyrrole rings generating chemically stable dicationic species. Protonation of the phenothiazine substituents required increased amounts of TFA, indicating phenothiazine as a weaker base in comparison to porphyrin. The distinct color change from brown to green observable with naked eye qualify the described ethynyl-MPP for potential applications as acid indicators for analytical purposes.

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
