# Peer review of "Ethyne Functionalized Meso-Phenothiazinyl-Phenyl-Porphyrins: Synthesis and Optical Properties of Free Base Versus Protonated Species"

_molecules, 2020, doi:10.3390/molecules25194546_

Round 1

Reviewer 1 Report

Recommendation: Publish after minor revisions.

Comments:

The article by Emese Gál,  Castelia Cristea  and co-workers reports on the preparation of functionalized porphyrins exhibiting particular light absorption properties in protonated form. This is logic continuation of previous studies  on meso-phenothiazinylporphyrin dyes published in Dye. Pigment. 2013, 99 (1), 144–153  and 2015, 123, 386–395.. The experiments are well planned and conducted and the results are clearly explained. I recommended publication in Molecules after addressing the following important points:

  1. Line 166. “In Figure 4 are shown the hyperporphyrin spectra of H2TPP2+ and H2MPP2+ .”

Please, revisit the conception of “hyperporphyrin” (ref 18, J. Phys. Chem. A 2011, 115, 10452; 2014, 118, 3605 and other). Why H2TPP2+ exibits a hyperporphyrin spectrum? Compare the spectra of H2MPP to the spectra of other porphyrins exhibiting this effect and add the corresponding discussion in the text. Why do you think that spectra of these compounds can be classified as “hyperporphyrin spectra” and how strong this effect in the studied compounds.

  1. Part Chemical synthesis. Please add the comment on the selective formation of only one (trans) of two isomers of disubstituted derivatives. If this is not the case, add a short discussion of this question.
  2. Line 70. “with extended conjugated π-electron systems”. There is a particular class of porphyrinoid derivatives named “extended porphyrins”. To distinguish the studied compounds from these derivatives I can recommended to use “Extended-conjugation π-electron systems” for description  of studied compounds. However, this is only my recommendation.

The following typos should be also corrected:

Line 81,89 “mixt”. Please, replace by mixed.

line 85 " 2:1 M ratio ". Please, replace M by molar.

line 163. “10-5M “. Please, add a space before M.

line 228. Please, use “h” instead of hours.

References: Please use CAS Source Index (CASSI) (https://cassi.cas.org/search.jsp) for journal abbreviation.

Reviewer 2 Report

Ethyne functionalized meso-phenothiazinyl-phenyl porphyrins: synthesis and optical properties of free  base versus protonated species.

This manuscript focuses on the Synthesis, structural characterization and photophysical properties for a series of new trans-A2B2 and A3B- type ethynyl functionalized meso-phenothiazinyl-phenyl porphyrin derivatives. The synthesized compounds were evaluated for their fluorescence emission properties. The changes recorded in the UV-vis absorption spectra in the presence of trifluoroacetic acid are consistent with the protonation of the two internal nitrogen atoms of the free-base porphyrin.

Protonation of the phenothiazine substituents required increased amounts of TFA and produced a distinct hypsochromic shift of the long wave absorption maxima. The DFT calculations of a porphyrin dication pointed out a saddle-distorted porphyrin ring as the ground-state geometry.

Using structural combination strategies, in order to increase the structural diversity of the porphyrin derivatives, based by their fluorescence emission properties, open perspectives for staining biological tissues in the  first NIR therapeutic window

The structures of all synthesized compounds were proved by analytical and spectroscopic data.

The Chapter Results and Discussion is of good scientific quality and the rich and instructive graphic realizes the understanding of the obtained results and of their significance. The experimental data is described appropriately and the manuscript needs no language and grammar corrections. The manuscript is written straight forward.

The purity, composition and structure of the compounds were determined using a variety of techniques, UV spectrometry and NMR, FT spectroscopy. The isolation, identification and characterization of , (H- and C-NMR) are described.  This is acceptable for organic compounds.

The heterocyclic derivatives are a valuable class of organic compounds, namely ligands of imidazoline receptors and chiral ligands for metal catalysis. The study is a meaningful suppliment  to the series of publications regarding the heterocyclic and macrocyclic compounds (with P, S, N atoms) related to natural products: synthesis, structural analysis and investigation of their biological activity, that  have been extensively studied because their important properties and applications, especially in biological activities, such as, anti-microbial, anti-proliferative (prostate cancer cells),  anti-cancer , anti-influenza A, neuraminidase (NA) , and with antioxidant activity. In Introduction the autors did not reflect any other field of another heterocyclic with the important applications as chiral ligands for metal catalyst or receptors especially in biological activities.

Examples of relevant publications are given below. It is recommended to the authors to cite these papers to give their introduction a wider base.

10-Heterocorroles: Ring-contracted porphyrinoids with fine-tuned aromatic and metal-binding properties, Sakow, D., Böker, B., Brandhorst, K. , Burghaus, O., Bröring, M., Angewandte Chemie - International Edition 2013, 52,  4912-4915

Chemistry of the l,3,5-Triaza-2-phosphinane-4,6-diones, Part V Synthesis of Phospho-ryl(III)(λ4P) and Thiophosphoryl(III)(λ4P) Derivatives of 1,3,5-Triaza-2-phosphinane-4,6- diones, Reactions with Ketones, Neda, I., Farkens, M., Fischer, A., Jones, P.G., Schmutzler, R., Zeitschrift fur Naturforschung - Section B Journal of Chemical Sciences, 1993, 48, 860-866.

Some references should be inserted.

In conclusion of my review

I recommend this manuscript for publication with minor revisions!

Reviewer 3 Report

In this manuscript, the authors developed a series of new ethynyl functionalized meso-phenothiazinyl-phenyl porphyrin derivatives with extended π-conjugated systems. Their photophysical properties were well investigated. The protonation at the level of pyrrole rings generated chemically stable dicationic species. The DFT calculations of a porphyrin dication pointed out a saddle-distorted porphyrin ring as the ground-state geometry. The manuscript can be accepted after minor revision.

  1. As mentioned “…displayed a high intensity absorption band with maxima in the UV spectral range (250-253 nm)”, but it is difficult to find corresponding UV spectral wavelength range in figure 1. Please provide the UV absorption spectrum with the full range.

  1. The authors believed that “minor bathochromic shift of the Soret band (up to 4 nm) upon peripheral functionalization of parent MPP (λ abs 420 nm) with ethynyl units”. However, why did some of peripheral functionalization compounds (2c, 2g,2i,2k) show blue shift or unchanged UV band?

  1. Figure 5 (a) shows normalized intensity, while the intensity is not normalized in fact.

  1. In terms of references, most literatures has been published for a long time, please cite some new references.

  1. There are many typos in the manuscript and the schemes (such as the reaction condition in scheme 1-3).

Reviewer 4 Report

This manuscript by Cristea et. al. is presented the synthesis of a series of trans-A2B2 and A3B-type “ethyne-functionalized” meso-phenothiazinyl-phenyl-porphyrin derivatives. The obtained porphyrin derivatives were fully characterized by their spectroscopic data and UV/vis absorption/emission spectra. The Synthetic methods for the porphyrin derivatives 2 are fundamentally the same as typical Adler-Longo mixt condensation reaction. In addition, the p-extended porphyrin derivatives described in this contribution is quite classic in the porphyrin chemistry. Thus, the synthetic methodology as well as fluorescent property might not be new enough for the publication in Molecules. In my opinion, I cannot recommend the acceptance of the present manuscript at this stage. I have found a few issues that, once addressed, will improve the manuscript:

  1. “meso” should be italic form. Additionally, “D” in the word “D2d symmetry” in page 2, line 63 should be corrected to italic form.
  2. There is no comparison with the previously reported ethynyl-substituted porphyrins in the UV/vis absorption/emission spectra and quantum yields. I request the explanation for the comparison between the obtained compounds and the related porphyrin derivatives.
  3. The synthetic schemes (Schemes 1 and 3) for the porphyrin derivatives 2 are very busy and complicated for readers. Since, for example, compounds 2a and 2b have different structures, it should be given different compound numbers such as 2 and 3. In addition, the reaction yields for all products in Schemes 1-3 should be added in the schemes.
  4. I think it is very important for the authors to comment in the experimental section on the Rf values for the obtained porphyrin derivatives 2 and how it must be separated.
  5. In the experimental section, the significant figures for the product yields (g) should be arranged.
  6. What about the fluorescent property of the obtained porphyrin derivatives 2 in the solid state and in the polar solvents?

Round 2

Reviewer 4 Report

I previously reviewed this manuscript in its original form. My original comments were all addressed in the revised version. Therefore, I suggest that the manuscript be accepted for publication in Molecules as is in its present form.